# Acid-Triggered Release of Eugenol and Fluoride by Desensitizing Macro- and Nanoparticles

**DOI:** 10.3390/jfb14010042

**Published:** 2023-01-11

**Authors:** Grigoriy Sereda, Abu Ahammadullah, Nisitha Wijewantha, Yulia Almiron Solano

**Affiliations:** Department of Chemistry, University of South Dakota, Vermillion, SD 57069, USA

**Keywords:** dentin, drug delivery, eugenol, fluoride, microparticles, calcium carbonate, casein, desensitizing, anti-bacterial, anti-cancer

## Abstract

The modern dentifrice industry needs non-toxic materials able to adhere to dentin, occlude dentinal tubules, hold pharmacons at the surface of dentin, and release them on demand to the location the tooth needs them most. Novel dental materials loaded with eugenol or fluoride-ions examined for the release of the pharmacon in an aqueous suspension efficiently adhere to the surface of human dentin and occlude dentinal tubules as evidenced by Scanning Electron Microscopy (SEM). Ultraviolet-visible (UV-vis) absorption spectroscopy and a fluoride-selective electrode quantified the release of pharmacons. The surface modification with casein stabilizes micro- and nanoparticles of calcium carbonate in aqueous suspensions, enabling their application in dentifrices. The ability of particles to hold and release eugenol depends on their morphology and composition, with the casein-coated calcium carbonate microspheres being the most acid-sensitive and most promising for dentifrice applications. The novel material releases fluoride under physiologically low pH, regardless of the presence of other ingredients of the artificial saliva, which sustains the bulk fluoride concentration comparable with most fluorinated toothpastes. Low pH-triggered release mechanisms selectively supply the drug to the areas that need it most, reducing the overall dose and ushering in a new type of targeted dentifrices.

## 1. Introduction

Dental diseases are one of the most common public health problems among all communities, affecting people throughout their lifetime and causing pain, discomfort, and disfigurement. According to the estimations from the Global Burden of Disease Study in 2016, oral diseases affected half of the world’s population (3.58 billion people) with dental caries (tooth decay) in permanent teeth [1]. More than half of children in the United States have at least one cavity or filling, which increases to 78% among teenagers [2]. The ultimate presence of these dental diseases is the damage of the tooth enamel (dental caries), inflammation of the periodontal gums (gingivitis), and the supporting periodontal tissues (periodontal diseases). Prevention, control, and responsive treatment are the most important actions for maintaining healthy teeth. Fluoride is a well-known and daily used remineralization agent delivered externally by toothpastes to prevent dental diseases such as dental caries. However, the major issue with fluoride-containing dentifrices is their short action time due to their dilution and clearance by saliva. Further, selective delivery of fluoride to the areas where the teeth need it the most would allow for a substantial decrease in the total amount of fluoride and alleviate potential issues with its toxicity. Bio-adhesive selective drug delivery systems will allow scientists to overcome these shortcomings. The toothpaste market is the largest section of the oral care market. In 2018, the toothpaste market was worth USD 26.1 billion and was poised to reach USD 37.0 billion by 2024 [3]. 

Hydroxyapatite (HA) [Ca_10_(PO_4_)_6_(OH)_2_] is the most common calcium phosphates phase, which has been substantially investigated in several biomedical applications due to its excellent biocompatibility, bioactivity, and chemical composition that is similar to natural bones and teeth [4,5]. HA particles are widely utilized as bone substitutions [6], dental materials [7], gene carriers [8], drug delivery carriers [9], ion exchangers [10], antibacterial agents [11], and catalysts [12]. Hydroxyapatite nanoparticles (HA NPs), along with other dentinal desensitizers, have been used in oral care products such as dentifrices and mouthwashes to temporarily [13] alleviate or eliminate tooth discomfort by preventing the opening of dentinal tubules that extend through the dentin to the pulp [14]. In a 2016 in vitro study, HA NPs demonstrated higher desensitizing potential than Novamin^®^ or Proargin^®^, which are both well-established desensitizing compositions [15]. HA NPs with bactericidal effects have been of great interest in minimizing the formation of dental cavities [16]. Preventing dental caries includes inhibiting or killing pathogenic oral bacteria, including *Streptococcus mutans*, in the lower layer of the oral biofilm. To date, several studies have determined the ability of various nanoparticles, including HA, to inhibit or kill these bacteria [17]. Adhesion of particles to dentin appears to be the initial step in occlusion, followed by other contributing mechanisms, such as particle aggregation, which triggers the deposition of more insoluble materials in the tubules and particle transport to the tooth surface [18]. The key advantage of HA NPs is their structural resemblance to the hydroxyapatite component of dentin and enamel, enabling their bioactivity and biocompatibility [19]. Earlier, we showed that hydroxyapatite nanoparticles can be delivered to the tooth surface by functionalized silk dental floss [20]. However, the potential of HA and other materials adhering to dentin to carry and release pharmaceuticals at the tooth surface remained unexplored. Further, more basic micro- and nanoparticles of vaterite or amorphous calcium carbonate with a better pH-buffering capability than HA have been so far unavailable for dental drug delivery applications due to their low thermodynamic stability [21].

Here, we report a study of the ability of novel and existing HA and calcium carbonate-based particles to carry and release eugenol and fluoride anions at the surface of dentin in a time- and pH-dependent manner. The low stability of this type of material at pH 5 is essential for the targeted drug delivery to the tooth areas that are mostly affected by the decay triggered by bacteriogenic acids. Eugenol was selected as the drug cargo because of its widespread use in the dental practice and its well-documented desensitizing [22], antibacterial [22], and anticancer [23] properties, while fluoride is a widely used pharmacon increasing the resistance of the hydroxyapatite component of teeth to the acidic challenge [24].

## 2. Materials and Methods

### 2.1. Preparation and Stabilization of Particles

#### 2.1.1. Calcium Carbonate Microspheres

Calcium acetate-based solution, CaCO_3_ (10 g, 0.1 mol), 50 mL of water, and acetic acid (12 g, 0.2 mol) were slowly mixed to control foaming. The mixture was diluted with water to 100 mL, kept overnight, and gravity filtered.

A 10 mL portion of 0.3 M NaHCO_3_ was added dropwise over 2 min to a vigorously stirred above-described calcium acetate-based solution (0.3 M, 10 mL). The mixture was not stirred for 5 min, then was stirred gently for 1 h, centrifuged at 8000 rpm for 5 min, washed with water (3 times by 2 mL, centrifuged at 8000 rpm for 5 min after each wash), redispersed in 1 mL of water, and dried on air. The yield was 100 mg. The particles were conjugated with casein according to procedure 2.2.

#### 2.1.2. Calcium Carbonate Nanospheres

Solution A. Polyethylene glycol methyl ether (Mw = 550, 20 mL) was added to 5 mL of 1 M aqueous calcium acetate. Solution B. Polyethylene glycol methyl ether (Mw = 550, 20 mL) was added to 5 mL of 1 M aqueous sodium bicarbonate. Calcium carbonate nanospheres were prepared according to the first step of the procedure [25], describing the fabrication of hollow hydroxyapatite nanoparticles by acidic etching of core-shell calcium carbonate–hydroxyapatite nanoparticles. Briefly, Solution A and Solution B were mixed at room temperature without stirring and incubated for 10 s. The resulting cloudy mixture was vigorously stirred for 4 h at room temperature and centrifuged at 11,000 rpm for 10 min. The separated white particles were washed with nanopure water (2X40 mL), centrifuged at 11,000 rpm for 10 min after each washing, and dried on air. The yield was 0.1 g.

#### 2.1.3. Calcium Carbonate/Hydroxyapatite Microplatelets 

Solution A. Polyethylene glycol methyl ether (Mw = 300, 20 mL) was added to 5 mL of 1 M aqueous calcium acetate. Solution B. Polyethylene glycol methyl ether (20 mL) was added to 5 mL of 1 M aqueous sodium bicarbonate. Solution A and Solution B were mixed at room temperature without stirring and incubated for 10 s. The resulting cloudy mixture was vigorously stirred (1200 rpm) with a magnetic stir bar for 4 h at room temperature. Next, the suspension was vigorously stirred (1200 rpm) with 2 mL of 0.1 M phosphoric acid for 17 h and centrifuged (11,000 rpm, 10 min). The separated white particles were washed with nanopure water (2X40 mL), centrifuged (11,000 rpm, 10 min) after each washing step, and dried on air. The yield was 0.241 g.

#### 2.1.4. Hydroxyapatite Particles 

The particles were prepared according to Huang et al. [26]. Briefly, a 100 mL portion of aqueous 0.05 M CaCl_2_ with PEG-300 (1.5% *w*/*v*), kept at room temperature for 12 h, was added with a burette at the flow rate of 1.6 mL/min into 100 mL of aqueous 0.03 M Na_2_HPO_4_ with constant stirring at 1000 rpm. The mixture was kept for 48 h at room temperature in a sealed vial and centrifuged at 11,000 rpm for 8 min. The product was washed with distilled water and ethanol three times, dried at 60 °C for 16 h, and calcined at 500 °C for 2 h. The yield was 0.26 g.

#### 2.1.5. Mesoporous Hydroxyapatite Particles 

The particles were synthesized based on the known procedure [27]. Briefly, polyethylene glycol methyl ether (20 mL) was added to 5 mL of 1 M aqueous calcium acetate to prepare Solution A. 20 mL of polyethylene glycol methyl ether was added to 5 mL of 1 M aqueous sodium bicarbonate to prepare Solution B. Solution A and Solution B were mixed at room temperature without stirring and were incubated for 10 s. The resulting cloudy mixture was stirred (1000 RPM) for 4 h at room temperature. The suspension was stirred (1000 rpm) with 2 mL of 0.01 M of phosphoric acid for 17 h and centrifuged (8000 rpm, 10 min). After each step, the separated white particles were washed with nanopure water (3X20 mL) and centrifuged (8000 rpm, 10 min). The precipitate was suspended in 10 mL of sodium acetate buffer (pH~6.0, 0.025 M) and stirred for 30 min. The particles were separated by centrifugation (8000 rpm, 10 min), washed three times with 20 mL of nanopure water, and dried overnight at ambient temperature to afford the final product. The yield was 0.85 g.

### 2.2. Conjugation of Particles with Casein 

A 1 mL portion of aqueous 1 mg/mL casein was added to 5 mg of particles. The mixture was mixed on a vortex for 5 min, placed in a Roto-Mini rotating mixer with 24 rpm for 30 min, and centrifuged at 8000 for 30 min. The supernatant was analyzed by the BCA protein assay [28] to determine the amount of casein not conjugated with the particles (See Appendix A). The amount of casein conjugated to each type of particles is presented in Appendix A.

Human teeth donated by Dr. Brower were extracted at the dental practice “Smiles For Siouxland” as a medically necessary procedure at the patients’ consent. The procedure was not performed with the purpose of performing any research. For the detailed experimental procedures for the preparation of dentin, application of particles to dentin, loading eugenol and fluoride-ions to the particles suspended in an aqueous solution or applied to dentin, see Appendix A.

## 3. Results

### 3.1. Preparation and Stabilization of Particles

#### 3.1.1. Calcium Carbonate Microspheres 

The SEM imaging of the 1–4 μm calcium carbonate microspheres prepared by our reproducible procedure shows their high porosity. The particles not stable in a 2 mg/mL aqueous suspension for more than 2 days were stabilized by pH 7.4 phosphate buffer saline (PBS) and much more efficiently—by a casein shell (7.2 µg casein/1 mg particles) (Figure 1).

#### 3.1.2. Calcium Carbonate Nanospheres 

The SEM imaging of the 50–100 nm calcium carbonate nanospheres prepared by our procedure shows their tendency to aggregate. Similarly to their microspherical counterpart, the particles are not stable in a 2 mg/mL aqueous suspension and were efficiently stabilized by a casein shell (5.6 µg casein/1 mg particles). (Figure 2). 

#### 3.1.3. Calcium Carbonate/Hydroxyapatite Microplatelets

The SEM imaging of calcium carbonate/hydroxyapatite core-shell microparticles prepared by our procedure reveals their morphology as 1–3 μm diameter platelets (Figure 3). 

The particles were stable in a 2 mg/mL aqueous suspension for at least 15 days. 

#### 3.1.4. Hydroxyapatite Particles 

Hydroxyapatite nanoneedles [26] and mesoporous microparticles [27] were synthesized according to known procedures. The particles’ size and morphology confirmed by Scanning Electron Microscopy (SEM) imaging (Figure 4) were consistent with those reported in the literature [26,27]. The X-Ray Diffraction (XRD) pattern of the hydroxyapatite nanoneedles (2T 26, 29, 32–34, 40, 46–54) was consistent with the presence of the (002), (211), (300), (202), (310), (222), (213), and (411) reflection planes of the hydroxyapatite phase according to the JCPDS files [29].

### 3.2. Adhesion of Particles to Dentin 

All synthesized particles have shown their ability to occlude dentinal tubules after application of an aqueous (100 mg/mL) paste followed by 1 min ultrasonication in water (Figure 5 and Figure 6).

### 3.3. Loading and Release of Eugenol by Aqueous Suspensions of Particles

Appendix A summarizes the capacity of bare and casein-coated particles for eugenol. The time profiles for the release of eugenol by casein-coated calcium carbonate microspheres measured for the pH values of 7.4 and 5.5 demonstrated the acid-triggered drug release. (Figure 7). The time profile for the release of eugenol by calcium carbonate/hydroxyapatite microplatelets at pH 7.4 was consistently erratic (Appendix A, black line). Coating particles with casein smoothens and slows the release of eugenol (Appendix A, blue line). The red line in Appendix A corresponds to the release of eugenol from casein-coated calcium carbonate/hydroxyapatite microplatelets. The time profiles for the release of eugenol by hydroxyapatite nanoneedles at pH 7.4 (Appendix A, black line), casein-coated hydroxyapatite nanoneedles at pH 7.4 (Appendix A, red line), and pH 5.5 (Appendix A, blue line). The time profiles for the release of eugenol by mesoporous hydroxyapatite microparticles at pH 7.4 (Appendix A, black line) and at pH 5.5 (Appendix A, red line). 

### 3.4. Release of Fluoride by Casein-Coated Calcium Carbonate Microspheres 

The time profiles for the release of fluoride by casein-coated calcium carbonate microspheres at pH 7.4 (Figure 8, black line), casein-coated calcium carbonate microspheres at pH 5.5 (Figure 8, red line), and artificial saliva at pH 6.9 (Figure 8, blue line).

### 3.5. Release of Eugenol by Particles Adhered to Dentin 

The time profiles for the release of eugenol by dentin treated with casein-coated calcium carbonate microspheres at pH 7.4 (Appendix A, red line) and pH 5.5 (Appendix A, blue line). Release from untreated dentin (Appendix A, black line). The time profiles for the release of eugenol by dentin treated with casein-coated calcium carbonate/hydroxyapatite microplatelets at pH 7.4 (Appendix A, red line) and pH 5.5 (Appendix A, blue line). Release from untreated dentin (Appendix A, black line). The time profiles for the release of eugenol by dentin treated with casein-coated hydroxyapatite nanoneedles at pH 7.4 (Appendix A, red line) and pH 5.5 (Appendix A, blue line). Release from untreated dentin (Appendix A, black line). The time profiles for the release of eugenol by dentin treated with mesoporous hydroxyapatite particles at pH 7.4 (Appendix A, black line) and pH 5.5 (Appendix A, red line). Release from untreated dentin (Appendix A, blue line).

## 4. Discussion

### 4.1. Preparation and Stabilization of Particles

#### 4.1.1. Calcium Carbonate Microspheres 

The calcium carbonate microspheres composed of thermodynamically unstable *vaterite* are known for their low stability in water because of their rearrangement to the more stable *calcite* [21]. Not surprisingly, reproducible synthesis of stable in the aqueous environment calcium carbonate particles of controlled morphology remains a challenge [21], which restricts a variety of their applications, such as that of carriers of biomolecules [30] and templates for microcapsules [31]. We developed a reproducible procedure for the precipitation of calcium carbonate microspheres from aqueous solutions of NaHCO_3_ and a calcium acetate-based solution prepared by reacting CaCO_3_ and aqueous acetic acid at the 1:2 molar ratio, and removed the unreacted CaCO_3_ by filtration. The pH value of the reaction mixture and evolving CO_2_ provide the perfect nucleation–crystal growth conditions for the calcium carbonate microspheres. 

Since the unstable *vaterite* rearranges to *calcite* through the process of Oswald ripening, which requires the diffusion of calcium ions into the solution, the stability of the *vaterite* particles can be improved by suppressing the diffusion. While the as-prepared particles (Figure 1A) are completely degraded after 2 days in water (Figure 1B), most of them retained their morphology even after 15 days in pH 7.4 PBS (Figure 1C) due to the formation of less soluble calcium phosphate on the surface of particles. Coating the particle with casein bound to the surface by its phosphoserine residues stabilizes the particles so efficiently that they remain intact even after 15 days in water (Figure 1D). This may also be due to a scaffolding effect of the casein coat.

#### 4.1.2. Calcium Carbonate Nanospheres

The stabilizing effect of casein on calcium carbonate spherical particles was found to be equally effective for both nanoparticles (50–100 nm) synthesized by a polymer-assisted method and microparticles (1–4 μm) (Figure 1 and Figure 2). While the calcium carbonate nanospheres completely degraded in an aqueous 2 mg/mL suspension after 7 days (Figure 2A), their casein-coated modification completely retained its morphology (Figure 2B). Both types of particles are stable as a dry powder even without casein, which confirms the role of the calcium ions diffusion in the *vaterite* rearrangement.

#### 4.1.3. Calcium Carbonate/Hydroxyapatite Microplatelets

To enhance the stabilizing effect of a phosphate layer on calcium carbonate particles, we prepared calcium carbonate/hydroxyapatite core-shell microparticles by treating intermediate calcium carbonate particles with aqueous phosphoric acid. The 1–4 μm particles did not undergo any visible changes after 15 days of exposure to water (Figure 3). Therefore, the hydroxyapatite shell coating prevented the rearrangement of calcium carbonate to the cubic *calcite* and produced platelet-like particles rather than spherical ones. 

#### 4.1.4. Hydroxyapatite Particles 

The hydroxyapatite particles of two distinctly different morphologies (nanoneedles and mesoporous spherical microparticles) were prepared to explore if the pH-dependent drug delivery functionality of the particles and their conjugates with casein can be tuned by their morphology. The needle-like particles were expected to better adhere to dentin and release the drug faster, while their mesoporous spherical counterparts were expected to hold larger amounts of the drug. 

### 4.2. Adhesion of Particles to Dentin 

To qualitatively evaluate the capability of particles to adhere to dentin and occlude its tubules by SEM imaging, it is critical to make sure that the dentinal tubules are open before applying particles (Figure 5A). This is achieved by removing any saw smears or debris from the surface of dentin by sonication with an aqueous citric acid followed by thorough rinsing with water. After the application of particles, all samples were sonicated in water for 30 s to remove loosely associated particles from the dentin. The casein-stabilized calcium carbonate microspheres efficiently adhere to dentin and occlude its tubules (Figure 5B). The most efficient visible occlusion was observed for mesoporous hydroxyapatite microparticles (Figure 5C), perhaps due to the mechanically rough surface of particles (Figure 4B). For hydroxyapatite nanoneedles and calcium carbonate/hydroxyapatite microplatelets, the casein coating does not apparently affect their occlusive properties (Figure 6).

### 4.3. Loading and Release of Eugenol by Aqueous Suspensions of Particles 

As shown in Appendix A, the morphology of particles does not substantially affect their capacity for eugenol, which tends to be about 15% higher for hydroxyapatite-core particles than for calcium carbonate-core particles. 

The casein-coated calcium carbonate microspheres slowly release eugenol into the solution at pH 7.4. Only 25% of the loaded eugenol was released after 24 h (Figure 7, red line). In contrast, at pH 5.5, 80% of eugenol was released after 8 h, sustaining the concentration of eugenol 67.2 µg/mL. This estimate of the maximum attainable concentration of the released eugenol is close to the concentration at which eugenol inhibits the proliferation of cancerous cells (82 μg/mL [23]) and is well below the concentration at which it starts to show toxicity for mammalian cells at prolonged exposures and bactericidal activity (164 μg/mL [32]). While the therapeutical application of eugenol itself is limited by the unfavorable balance of its therapeutical activity and toxicity [22], the local concentrations of eugenol in a confined environment (inside dentinal tubules or underneath biofilms) can be expected to exceed these levels at least for a brief time, which underlines the importance of targeted delivery of eugenol by novel materials. Only 15% of eugenol was released from the particles after 8 h at pH 7.4. About a 5× increase in the drug release triggered by acid is crucial for the targeted delivery of eugenol to the areas of teeth exposed to the bacteriogenic acids starting the process of tooth decay. 

The calcium carbonate/hydroxyapatite core-shell microplatelets have shown a similar capacity for eugenol as casein-coated calcium carbonate microspheres. Interestingly, the non-coated microplatelets released eugenol erratically (Appendix A, black line), likely due to the heterogeneous nature of their surface and the re-adsorption of eugenol from the solution. The casein coating sealed the eugenol inside the particles, slowed down (0.3× after 8 h), and smoothed its release (Appendix A, blue line). The casein-coated core-shell microplatelets were as responsive to the acid-triggered release (5× increase after 8 h) as calcium carbonate microspheres.

Switching the hydroxyapatite morphology from microplatelets to nanoneedles increases the particles’ capacity for eugenol by ~30%, but decreases their ability to hold eugenol, probably, because of the prevalence of its surface adsorption versus diffusion inside the particles. After 8 h, 90% of eugenol was released (Appendix A, black line). Similarly to other types of particles, coating with casein substantially increased the retention of eugenol by the particles: only 15% was released after 8 h, and 75% of eugenol remained inside the particles even after 24 h (Appendix A, red line). However, acidification to pH 5.5 exerted just a moderate 1.7× increase in the release after 8 h (Appendix A, blue line). Therefore, casein-coated hydroxyapatite nanoneedles hold more eugenol than calcium carbonate-based microspheres and microplatelets for 24 h, but they are much less sensitive to the acid-triggered release. 

As expected, mesoporous hydroxyapatite particles were able to hold eugenol better (45% of eugenol released after 8 h, Appendix A) than hydroxyapatite nanoneedles (90% of eugenol released after 8 h, Appendix A). Similarly to hydroxyapatite nanoneedles, mesoporous hydroxyapatite particles have shown moderate sensitivity to the acid-triggered release (1.4× increase of the release upon switching pH from 7.4 to 5.5, Appendix A). The observed tendency of calcium carbonate-based particles to be more sensitive to the acid-triggered drug release than hydroxyapatite particles is not surprising considering the higher solubility of calcium carbonate than hydroxyapatite at pH 5.5.

Coating with casein played a unique role for each type of particles. The casein layer stabilized the calcium carbonate microspheres, smoothed the release of eugenol by calcium carbonate/hydroxyapatite microplatelets, enabled the retention of eugenol by hydroxyapatite nanoneedles, and was not needed for mesoporous hydroxyapatite particles.

### 4.4. Release of Fluoride by Casein-Coated Calcium Carbonate Microspheres 

The casein-coated calcium carbonate microspheres as the best candidate for the acid-triggered release of eugenol in dentifrice applications were explored for their ability to release the fluoride-ion as well. The fluoride release time profiles were collected at three values of pH: 7.4 (Phosphate Buffered Saline (PBS) buffer), 6.9 (artificial saliva Biotene^®)^, and 5.5 (PBS buffer). Figure 8 shows a clear trend of the substantially accelerated release of eugenol as the pH value decreases, sustaining the maximum cumulative fluoride concentration of 850 ppm, which is comparable with the 1000–1500 ppm for most fluorinated toothpaste. This trend is not affected by the ingredients of artificial saliva, which speaks to the value of the particles for dentifrice applications as a source of both eugenol and fluoride.

### 4.5. Release of Eugenol by Particles Adhered to Dentin 

While dentin itself can absorb eugenol, the treatment with eugenol-loaded particles increased its capacity 3× for casein-coated hydroxyapatite nanoneedles and 5-7× for other types of particles (Appendix A). Similarly to the aqueous suspensions, acidification from pH 7.4 to pH 5.5 roughly doubled the eugenol release in 2–4 h by casein-coated calcium carbonate microspheres and casein-coated calcium carbonate/hydroxyapatite microplatelets. For the particles with the hydroxyapatite core, the pH effect on the release was less pronounced than for the particles with the calcium carbonate core due to the higher basicity of calcium carbonate. The overall amount of eugenol released by hydroxyapatite-core particles at pH 7.4 was about 15% less than the amount released at pH 5.5. The bulk concentrations of eugenol released by the particles on dentin (5–10 μg/mL) are much lower than for eugenol released in an aqueous suspension and are well below the biologically active concentrations. Therefore, the particles hold eugenol at the surface of dentin and locally release it at lower pH values while eliminating the risk of systemic exposure to eugenol, including its ingestion. The evaluation of local concentrations of eugenol will be performed by using a dentin-embedded microfluidic device [33].

## 5. Conclusions

The surface modification with casein stabilizes micro- and nanoparticles of calcium carbonate in aqueous suspensions and, therefore, enables their application in dentifrices (Figure 9). A series of nano- and microparticles based on calcium carbonate, calcium hydroxyapatite, and casein hold antibacterial, desensitizing, and anticancer eugenol, adhere to dentin, and release eugenol in an acid-triggered manner, sustaining the bulk eugenol concentration in a solution comparable with its anti-proliferation levels, but below the antibacterial and mammalian toxicity levels. The adhesion of particles to dentin enables the tooth surface to hold and release eugenol. The ability of particles to hold and release eugenol depends on their morphology and composition, with the casein-coated calcium carbonate microspheres being the most acid-sensitive and most promising for dentifrice applications. The casein-coated and fluoride-loaded calcium carbonate microspheres release fluoride in a pH-dependent manner regardless of the presence of other ingredients of the artificial saliva, sustaining the bulk fluoride concentration comparable with most fluorinated toothpastes.

Our findings are opening the door to a new class of desensitizing and therapeutic dentifrices and compositions.

## Figures and Tables

**Figure 1 jfb-14-00042-f001:**
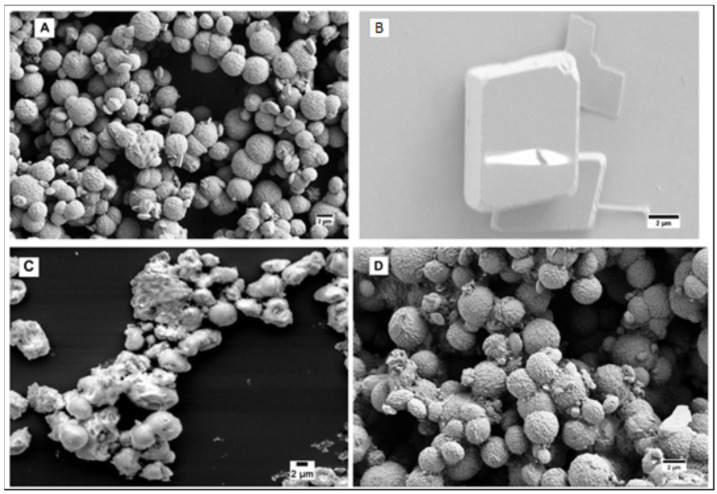
(**A**) As prepared calcium carbonate microspheres; (**B**) Calcium carbonate microspheres after 2 days in water; (**C**) Calcium carbonate microspheres after 15 days in pH 7.4 PBS; (**D**) Casein-coated calcium carbonate microspheres after 15 days in water.

**Figure 2 jfb-14-00042-f002:**
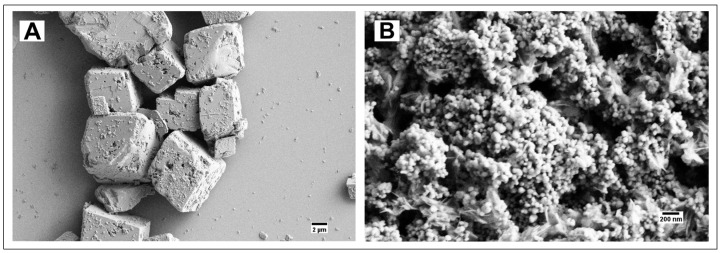
(**A**) Calcium carbonate nanospheres after 7 days in water; (**B**) Casein-coated calcium carbonate nanospheres after 7 days in water.

**Figure 3 jfb-14-00042-f003:**
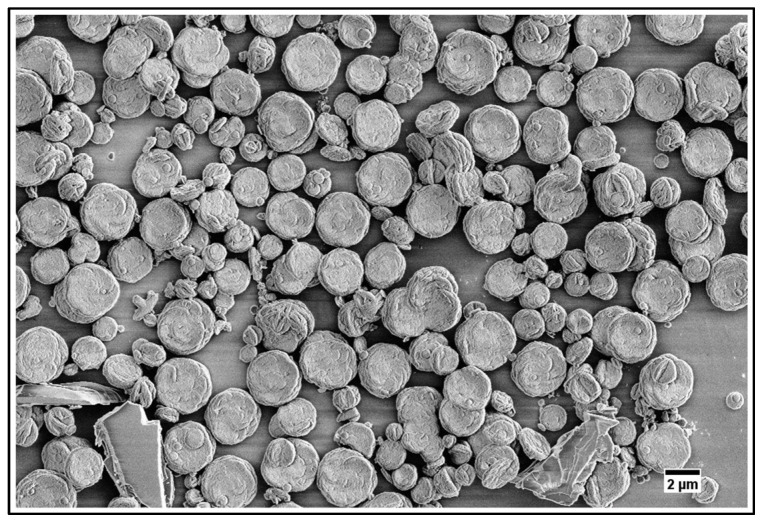
Calcium carbonate/hydroxyapatite core-shell microplatelets.

**Figure 4 jfb-14-00042-f004:**
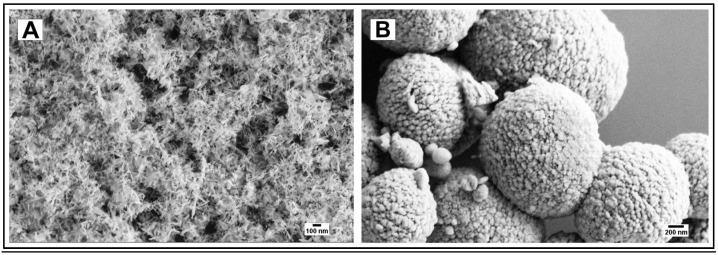
(**A**) Hydroxyapatite nanoneedles; (**B**) Hydroxyapatite mesoporous microparticles.

**Figure 5 jfb-14-00042-f005:**
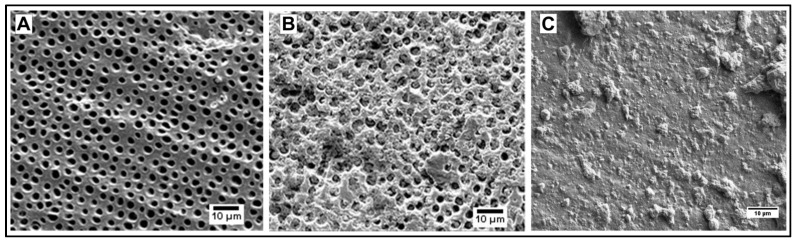
(**A**) Dentin before application of particles; (**B**) Dentin occluded by casein-coated calcium carbonate microspheres; (**C**) Dentin occluded by mesoporous hydroxyapatite microparticles.

**Figure 6 jfb-14-00042-f006:**
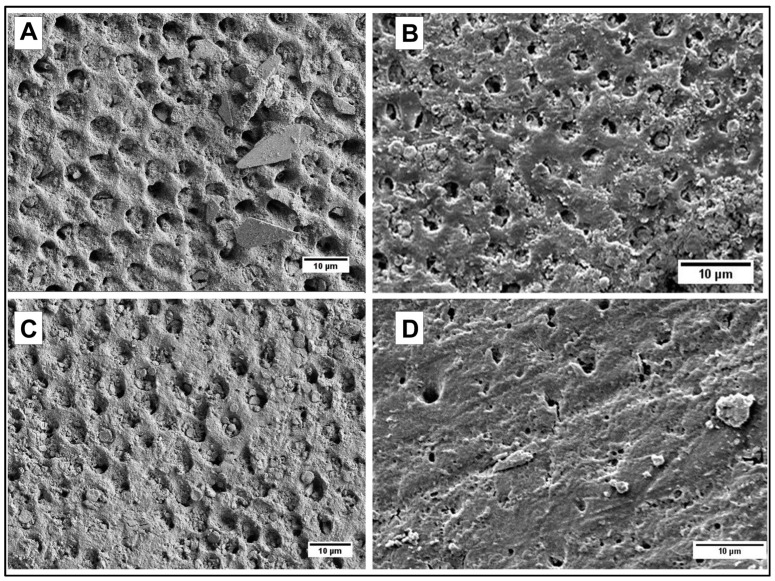
(**A**) Dentin occluded by hydroxyapatite nanoneedles; (**B**) Dentin occluded by casein-coated hydroxyapatite nanoneedles; (**C**) Dentin occluded by calcium carbonate/hydroxyapatite microplatelets; (**D**) Dentin occluded by casein-coated calcium carbonate/hydroxyapatite microplatelets.

**Figure 7 jfb-14-00042-f007:**
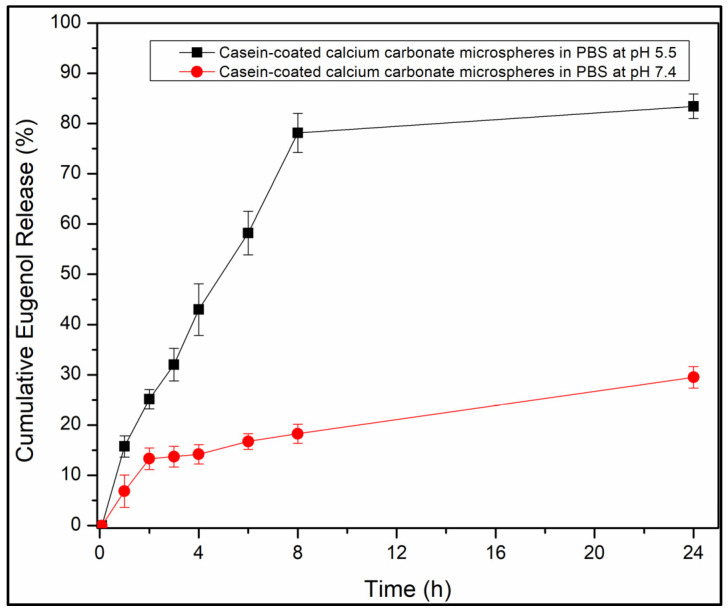
Cumulative eugenol release profiles from casein-coated calcium carbonate microspheres in phosphate buffer saline at pH 7.4 (red line) and pH 5.5 (black line).

**Figure 8 jfb-14-00042-f008:**
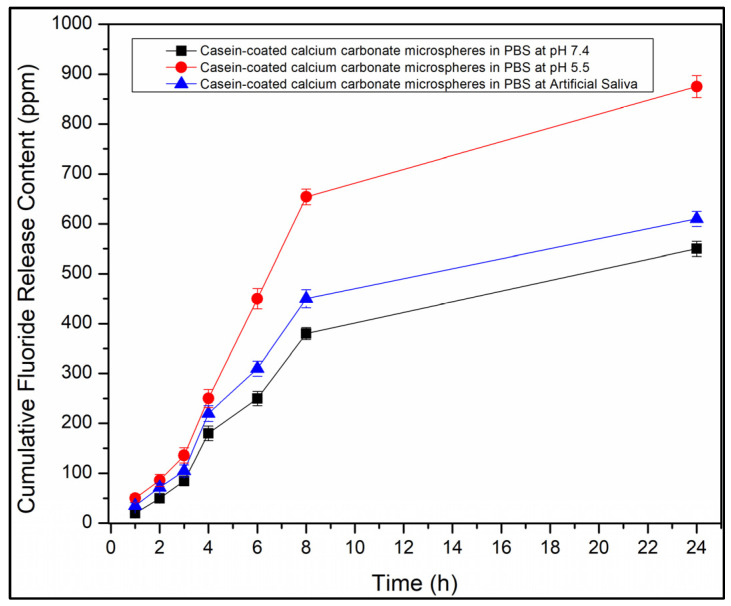
Cumulative F^-^-release profiles from casein-coated calcium carbonate microspheres in phosphate buffer saline at pH 7.4 (black line). Cumulative eugenol release profile from casein-coated calcium carbonate microspheres in phosphate buffer saline at pH 5.5 (red line) and artificial saliva at pH 6.9 (blue line).

**Figure 9 jfb-14-00042-f009:**
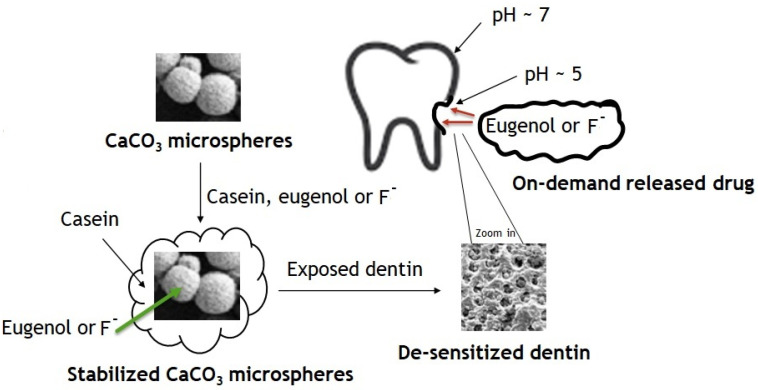
Synthesis of modified calcium carbonate microspheres and their application for acid-triggered drug delivery.

## Data Availability

Not applicable.

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
