# Peer review of "Acid-Triggered Release of Eugenol and Fluoride by Desensitizing Macro- and Nanoparticles"

_jfb, 2023, doi:10.3390/jfb14010042_

Round 1
Reviewer 1 Report
It is not easy to read so much and too much information in SI.
Quantification of eugenol and fluro release are unclear
XRD results should also be included
SEM images are too many and useless sometimes
Author Response
It is not easy to read so much and too much information in SI.
Figures 1S and 5S have been moved from SI to the main manuscript.
Quantification of eugenol and fluro release are unclear
Quantification of eugenol release is now demonstrated by figure 7 and quantification of fluoride release is now demonstrated by figure 8 in the main manuscript.
XRD results should also be included
The XRD results are now presented in section 3.1.4. of the manuscript.
SEM images are too many and useless sometimes
Figure 3b (an SEM image) is now removed as it describes an unchanged material after a prolonged exposure to water. This result is better presented by text.
Reviewer 2 Report
Acid-triggered Release of Eugenol and Fluoride by Desensitizing Macro- and Nanoparticles
While the work is overall carried out well and the review support the conclusion, there are several issues that need attention and upon addressing those issues the paper can be accepted
1- Firstly, there are numerous typos (overtyping) throughout the manuscript, all requiring attention (the abstract has such errors). There are several grammatical errors that needed to be corrected. I urge the authors to thoroughly go through the entire manuscript and check every line for spelling, grammar, or sentence construction-related errors as without these measures the account is unreadable.
2- Explain each word for first time in the beginning and used its abbreviation after that [eg: hydroxyapatite (HA)]
3- Proper conclusion outcome of all items to be presented in the manuscript

Author Response
While the work is overall carried out well and the review support the conclusion, there are several issues that need attention and upon addressing those issues the paper can be accepted
1- Firstly, there are numerous typos (overtyping) throughout the manuscript, all requiring attention (the abstract has such errors). There are several grammatical errors that needed to be corrected. I urge the authors to thoroughly go through the entire manuscript and check every line for spelling, grammar, or sentence construction-related errors as without these measures the account is unreadable.
The revised version of the manuscript is carefully checked for typos and grammatical errors both manually and by the Word-implemented grammar and spelling checker.
2- Explain each word for first time in the beginning and used its abbreviation after that [eg: hydroxyapatite (HA)]
Done, specifically, for SEM, UV-vis, XRD, PBS.
3- Proper conclusion outcome of all items to be presented in the manuscript
The conclusions section is now supplemented by figure 9 summarizing the proper outcome of all items presented in the manuscript. A new sentence underlining the practical relevance of the work has also been added to the conclusions section.
Reviewer 3 Report
Dear authors, thank you for the submission. This study is well written and presents a significant scientific importance. However, few points need to be improved, as follow:
Abstract: abstract structured is unsatisfactory. I recommend improving.
The Materials and Methods topic is well described; however, I recommend considering making graphs or diagrams demonstrating particle preparation, in order to facilitate or complement the reader’s understanding.
Simple method of study, using SEM imagens alone, but which provide significant information. The data is supported by the imagens; however, all graphics are supplemental material. Consider inserting some more important graphics for the body of the manuscript.
Author Response
Dear authors, thank you for the submission. This study is well written and presents a significant scientific importance. However, few points need to be improved, as follow:
Abstract: abstract structured is unsatisfactory. I recommend improving.
In the revised manuscript, the abstract is structured according to the journal’s guidelines for authors.
The Materials and Methods topic is well described; however, I recommend considering making graphs or diagrams demonstrating particle preparation, in order to facilitate or complement the reader’s understanding.
The particles were prepared by a variety of methods. The general strategy of synthesis and application is now demonstrated by figure 9 added to the conclusions section.
Simple method of study, using SEM imagens alone, but which provide significant information. The data is supported by the imagens; however, all graphics are supplemental material. Consider inserting some more important graphics for the body of the manuscript.
Quantification of eugenol release is now demonstrated by figure 7 and quantification of fluoride release is now demonstrated by figure 8 in the main manuscript. The figures have been moved from the Supplementary section.
Round 2
Reviewer 1 Report
ok in the present form
Author Response
Thank you.
Reviewer 2 Report
Dear editor,
The authors replied to all comments.
best regards
Sanaa
Author Response
Thank you.
Reviewer 3 Report
Dear author, I have no more questions. Thank you.
Author Response
Thank you.